# Age and remission induction therapy for acute myeloid leukemia: An analysis of data from the Korean acute myeloid leukemia registry

**Kwai Han Yoo** [1], **Hyeoung-Joon Kim**[2], **Yoo Hong Min**[3], **Dae-Sik Hong**[4], **Won Sik Lee**[5], **Hee-Je Kim**[6], **Ho-Jin Shin**[7], **Yong Park**[8], **Je-Hwan Lee**[9], **Hawk Kim**[1] *

1 Gachon University Gil Medical Center, Gachon University College of Medicine, Incheon, Korea, 2 Chonnam National University Hwasun Hospital, Chonnam National University College of Medicine, Gwangju, Korea, 3 Severance Hospital, Yonsei University College of Medicine, Seoul, Korea, 4 Soonchunhyang University Hospital, Soonchunhyang University College of Medicine, Bucheon, Korea, 5 Inje University Busan Paik Hospital, Inje University College of Medicine, Busan, Korea, 6 Catholic Hematology Hospital, Seoul St. Mary's Hospital, College of Medicine, The Catholic University of Korea, Seoul, Korea, 7 Pusan National University Hospital, Pusan University College of Medicine, Busan, Korea, 8 Korea University Anam Hospital, Korea University College of Medicine, Seoul, Korea, 9 Asan Medical Center, University of Ulsan College of Medicine, Seoul, Korea

* kimhawkmd@gmail.com

**Data Availability Statement:** All relevant data are within the paper and its Supporting information files.

## Abstract

### Objective

The clinical characteristics and therapeutic strategy in acute myeloid leukemia (AML) are influenced by patients' age. We evaluated the impact of age on remission induction therapy for AML.

### Methods

We retrospectively analyzed 3,011 adult AML patients identified from a nationwide database between January 2007 and December 2011.

### Results

Three hundred twenty-nine (10.9%) acute promyelocytic leukemia (APL) and 2,682 (89.1%) non-APL patients were analyzed. The median age was 51 years and 55% of patients were male. Six hundred twenty-three patients (21%) were at favorable risk, 1522 (51%) were at intermediate risk, and 743 (25%) were at poor risk. As the age increased, the proportion of those at favorable risk and who received induction chemotherapy decreased. After induction therapy, complete response (CR) was achieved in 81.5% (243/298) of APL and 62.4% (1,409/2,258) of non-APL patients; these rates decreased as the age increased, with an obvious decrement in those older than 60 years. The median overall survival of non-APL patients was 18.7 months, while that of APL patients was not reached, with a 75% five-year survival rate.

**Funding:** HK was supported by the Gachon University research fund of 2017 (GCU-2017-5257). This study was also supported by The Korean Society of Hematology AML/MDS Working Party (AMLMDSWP-15-R-02). The funder had no role in study design, data collection and analysis, decision to publish, or preparation of the manuscript. Gachon University (https://www.gachon.ac.kr) Korean Society of Hematology AML/MDS Working Party (https://www.hematology.or.kr/research_member/main.html?comm_code=com_01).

**Competing interests:** The authors have declared that no competing interests exist.

## Conclusions

Age impacts both the biology and clinical outcomes of AML patients. Further studies should confirm the role of induction remission chemotherapy by age group.

## Introduction

Acute myeloid leukemia (AML), characterized by the clonal expansion of myeloid blasts resulting from somatic mutations in primitive multipotential hematopoietic cells, is the most common acute leukemia in adults [1]. The median age of patients with AML at the time of diagnosis was reported to be around 70 years in the Western population and the prevalence rate of AML is strongly associated with age [2, 3]. The clinical and biologic characteristics of patients with AML are closely linked with the aging process and the management and treatment outcomes are significantly influenced by the patient's age [4, 5]. The initial treatment strategy for AML is established by considering patients' age and performance status. Those younger than 60 years are generally treated with remission induction chemotherapy based on a backbone of cytarabine plus an anthracycline [6]. However, the treatment of older adults with AML encounters two major obstacles: therapeutic resistance of the disease and patients' intolerance to intensive chemotherapy [7]. Thus, the rate of remission induction therapy performed in elderly AML patients is reduced and other therapeutic options such as hypomethylating agents, low-dose cytarabine, or best supportive care with oral cytostatic drugs may instead be introduced [8]. Here, we explored the incidence and disease characteristics of AML by age in the Korean population and evaluated the impact of age on the use of remission induction therapy and treatment outcomes among patients with AML.

## Methods and methods

### Patients and data collection

This study was a multicenter, retrospective, longitudinal cohort study using data sourced from the nationwide Korean AML Registry. The data were provided by the AML/myelodysplastic syndrome working party of the Korean Society of Hematology (KSH). To create this registry, a total of 3,041 AML patients were enrolled from 28 institutions between January 2007 and December 2011. Their medical records were reviewed for the baseline characteristics, diagnosis, cytogenetic risk stratification, treatments, and survival outcomes. The registry includes patients aged 14 years or older; however, we analyzed only adult AML patients who were 18 years or older.

   The diagnosis of AML was made by confirmation of a 20% or greater concentration of leukemic blasts in the bone marrow or peripheral blood according to the World Health Organization (WHO) classification system. Risk stratification in this study was performed according to the Cancer and Leukemia Group B (CALGB) stratification scheme using karyotyping and fluorescence in situ hybridization cytogenetic analysis, which included core binding factor; normal, complex and monosomal karyotypes; and other chromosomal changes [9]. A complex karyotype was defined as any karyotype with at least three chromosome aberrations, regardless of their type and the individual chromosomes involved. A monosomal karyotype was defined as at least two autosomal monosomies or one single autosomal monosomy with one or more structural cytogenetic abnormalities, excluding acute promyelocytic leukemia (APL) and core binding factor AML. However, molecular prognostic markers such as FLT3, NPM1, CEBPA,

or KIT mutation were not considered in this study for stratifying the risk groups since the appropriate data were not available from every institution during the period of time covered by the registry. Age groups were categorized as follows: 18 to 20 years, 21 to 80 years divided into individual decades, and 81 years or older. We stratified the patients according to the presence of acute promyelocytic leukemia (APL) or not, receiving induction chemotherapy or not, and undergoing a response evaluation after induction therapy or not. Induction chemotherapy was categorized as adherence to regimens containing cytarabine plus idarubicin (AI), cytarabine plus daunorubicin (AD), or others.

This study was conducted according to the Declaration of Helsinki and was approved by the institutional review board of all participating hospitals (Gachon University Gil Medical Center, Chonnam National University Hwasun Hospital, Yonsei Cancer Center, Soonchunhyang University Hospital, Inje University Busan Paik Hospital, Seoul St. Mary's Hospital, Pusan National University Hospital, Korea University Anam Hospital, and Asan Medical Center). In light of the retrospective nature of the study, the need for informed consent was waived.

## Statistics

Differences between groups were assessed using a Student's t-test for continuous variables, while the comparison of dichotomous or categorical variables was performed using Pearson's chi-squared test or Fisher's exact test. Overall survival (OS) was measured from the date of diagnosis until death from any cause, with observations censored for patients who remained alive at the last date of follow-up. Distributions of OS were estimated using the Kaplan–Meier method. Comparisons of OS between groups were made using the log-rank test. All p-values were two-tailed and $p < 0.05$ was considered to be statistically significant. All data were analyzed using the Statistical Package for the Social Sciences version 23.0 software program (IBM Corporation, Armonk, NY, USA).

## Results

### Patient characteristics

Of the 3,041 patients in the registry, 30 patients were excluded (including one patient who was misdiagnosed, one patient who was missing age data, and 28 patients younger than 18 years old); finally, 3,011 adult AML patients were identified for inclusion in the final analysis (Fig 1). Among these, 329 patients (10.9%) were diagnosed with APL and 2,682 patients (89.1%) were non-APL cases. Further, 298 patients (90.6%) with APL and 2,258 patients (84.2%) without APL had received induction remission chemotherapy. Among the non-APL patients with a history of induction therapy and who were assessable to discern the treatment response (n = 1,938), 1,155 (59.6%) patients had received AI and 225 (11.6%) patients had been treated with AD as induction remission therapy.

The baseline characteristics by age groups are given in Table 1. The median age of all study participants was 51 years (range: 18–89 years) and 55% of patients were male. According to cytogenetic risk stratification, 21% of patients (n = 623) were at favorable risk, 51% (n = 1,522) were at intermediate risk, and 25% (n = 743) were at poor risk. The median values of blast percentages at the time of diagnosis in the peripheral blood and bone marrow were 36% and 65%, respectively. The number of patients and the composition of cytogenetic risk by age group are presented in Fig 2. The number of patients increased by age group from the second to the fourth decade of life, while more than 600 patients were enrolled in each decade age group between 41 and 70 years. After the plateau between 41 and 70 years, the number of patients

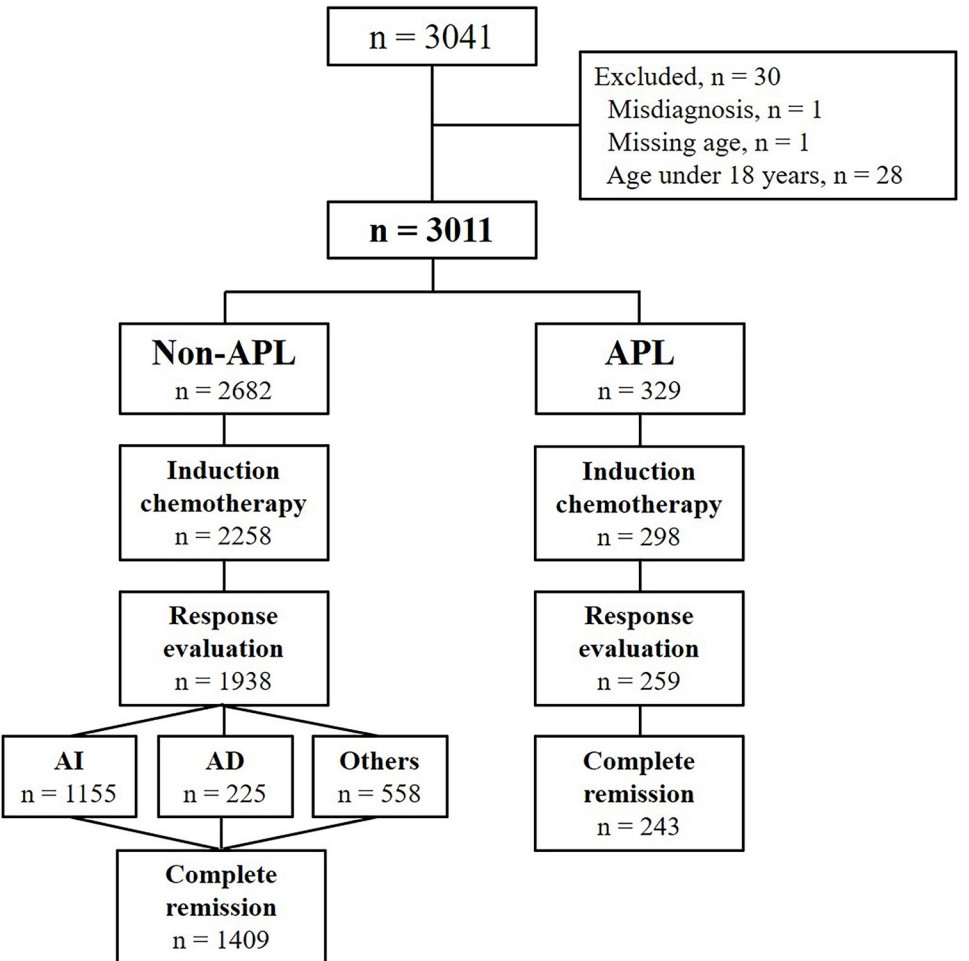

**Fig 1. Flow diagram.** APL: acute promyelocytic leukemia, AI: cytarabine plus idarubicin, AD: cytarabine plus daunorubicin.

decreased by less than 300 in the age group from 71 to 80, while just 39 patients older than 80 years were included in the registry (Fig 2A).

Meanwhile, cytogenetic risk-group stratification based on karyotype showed strikingly different tendencies according to the age group under consideration. Particularly, the favorable-risk group included a majority of patients in the age group of 18 to 20 years and 27% to 29% of patients in the decade age groups between 21 and 50 years. However, the number of patients at favorable risk started to diminish beginning at 50 years of age and only 6% of patients older than 70 years of age were in the favorable-risk group. On the contrary, both the intermediate-and poor-risk groups showed trends of gradual population increases after 50 years (Fig 2B).

## Treatment outcome by age and cytogenetic risk

The relationship between age and induction chemotherapy is shown in Fig 3. Among patients younger than 60 years, more than 90% had received induction chemotherapy. Conversely, the rate of induction progressively decreased among those older than 60 years, with less than 50% of patients older than 70 years having received induction therapy (Fig 3A). After induction remission therapy, treatment response data were assessable for 259 patients (86.9%) with APL

**Table 1. Baseline characteristics.**

| Age group (years) | 18–20 | 21–30 | 31–40 | 41–50 | 51–60 | 61–70 | 71–80 | $\geq 81$ | Total |
|---|---|---|---|---|---|---|---|---|---|
| **No. of patients** | 75 | 326 | 444 | 612 | 623 | 601 | 291 | 39 | 3011 |
| **Sex, male** | 45 (60%) | 171 (53%) | 244 (55%) | 302 (49%) | 369 (59%) | 348 (58%) | 170 (58%) | 17 (44%) | 1666 (55%) |
| **Non-APL** | 64 (85%) | 286 (88%) | 377 (85%) | 521 (85%) | 550 (88%) | 562 (94%) | 283 (97%) | 39 (100%) | 2682 (89%) |
| **Cytogenetic risk** | | | | | | | | | |
| Favorable | 33 | 96 | 119 | 174 | 114 | 68 | 17 | 2 | 623 (21%) |
| Intermediate | 22 | 128 | 196 | 304 | 340 | 337 | 174 | 21 | 1522 (51%) |
| Poor | 18 | 82 | 114 | 126 | 145 | 171 | 75 | 12 | 743 (25%) |
| Unknown | 2 | 20 | 15 | 8 | 24 | 25 | 25 | 4 | 123 |
| **Laboratory findings at diagnosis** | | | | | | | | | |
| Hb (g/dL) | 8.7 | 8.7 | 8.4 | 8.2 | 8.6 | 8.5 | 8.2 | 8.5 | 8.4 |
| Platelets ($10^3$/mm$^3$) | 41 | 48 | 46 | 48 | 52 | 57 | 51 | 66 | 50 |
| PB blasts (%) | 59 | 48 | 38 | 42 | 29 | 31 | 24 | 32 | 36 |
| BM blasts (%) | 76 | 74 | 70 | 66 | 64 | 60 | 56 | 58 | 65 |

APL: acute promyelocytic leukemia, BM: bone marrow, PB: peripheral blood.

and 1,938 patients (85.8%) with non-APL. Finally, complete response (CR) was achieved in 81.5% (243/298) of APL and 62.4% (1409/2258) of non-APL patients who received induction therapy, respectively. The CR rates were observed to have decreased as age increased, with an obvious decrement existing among those older than 60 years (Fig 3B). Age variations with regard to induction remission chemotherapy and the CR rate in APL and non-APL patients are given in Fig 3C and 3D, respectively.

The median OS of all patients and non-APL patients were 23.3 months [95% confidence interval (CI): 19.9–26.6] and 18.7 months (95% CI: 16.7–20.7), respectively. The median OS among APL patients was not reached, with a 75% five-year survival rate. The OS curves of the entire population by age group and cytogenetic risk group are given in Fig 4. The survival curves were well-separated from one another according to age group (Fig 4A) and cytogenetic risk (Fig 4B), respectively. Differences in the OS by age and cytogenetic risk group were observed among non-APL patients but not among APL patients (S1 Fig).

### Treatment outcome by induction regimen

Focusing on non-APL patients who received induction remission therapy by AI (n = 1,155) and AD (n = 225) regimens with assessable treatment-response data, CR rates were 74% in patients who received the AI regimen and 72% among those who received the AD regimen, respectively (p = 0.528). In the comparison of CR rates determined by induction regimen according to age, no statistically significant difference between the AI and AD regimens in any age group was apparent (Table 2).

The median OS among patients who received the AD regimen (39.0 months) was longer than that among patients who received the AI regimen (24.9 months, 95% CI: 20.3–29.4; p = 0.029) (Fig 5A). When dividing the patients into two groups using the age of 60 years as a cutoff, however, the difference in OS according to the treatment regimen was not significant in either age group (Fig 5B and 5C). Considering cytogenetic risk, no significant difference in OS according to the treatment regimen in the favorable- or poor-risk group was apparent (Fig 6A and 6C). However, patients receiving the AD regimen (median was not reached) showed improved OS as compared with those receiving the AI regimen (24 months, 95% CI: 18.9–29.1) in the intermediate-risk group (p = 0.001) (Fig 6B). Non-APL patients who had

(A)

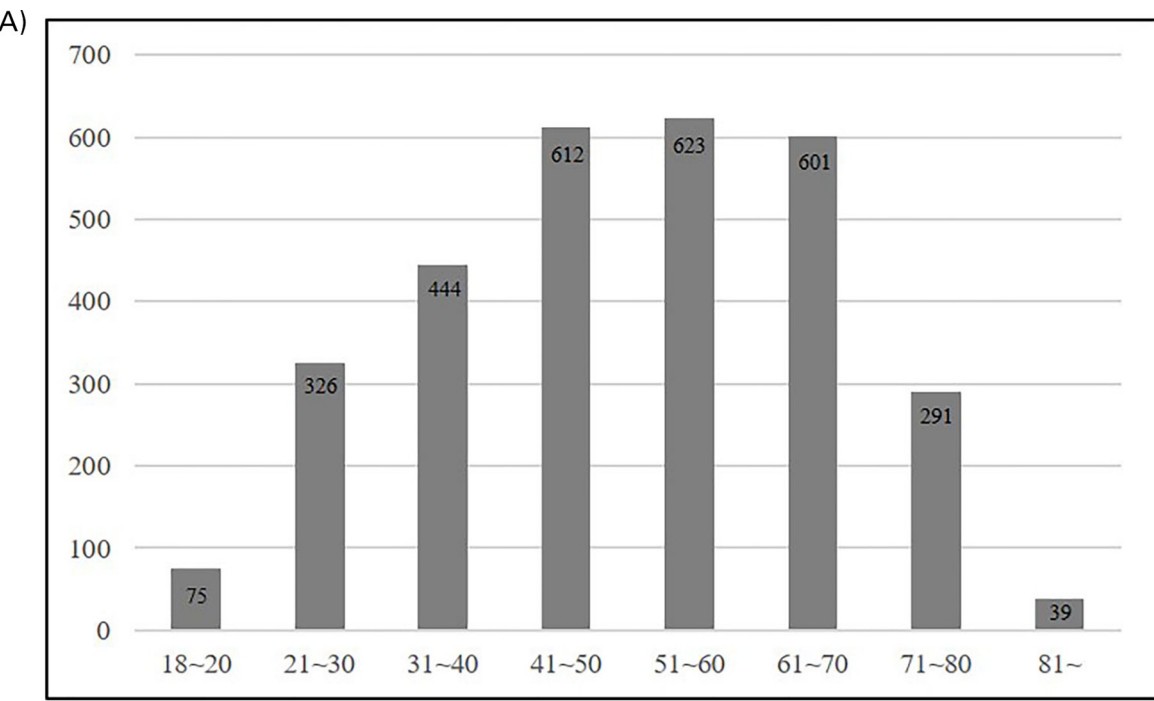

(B)

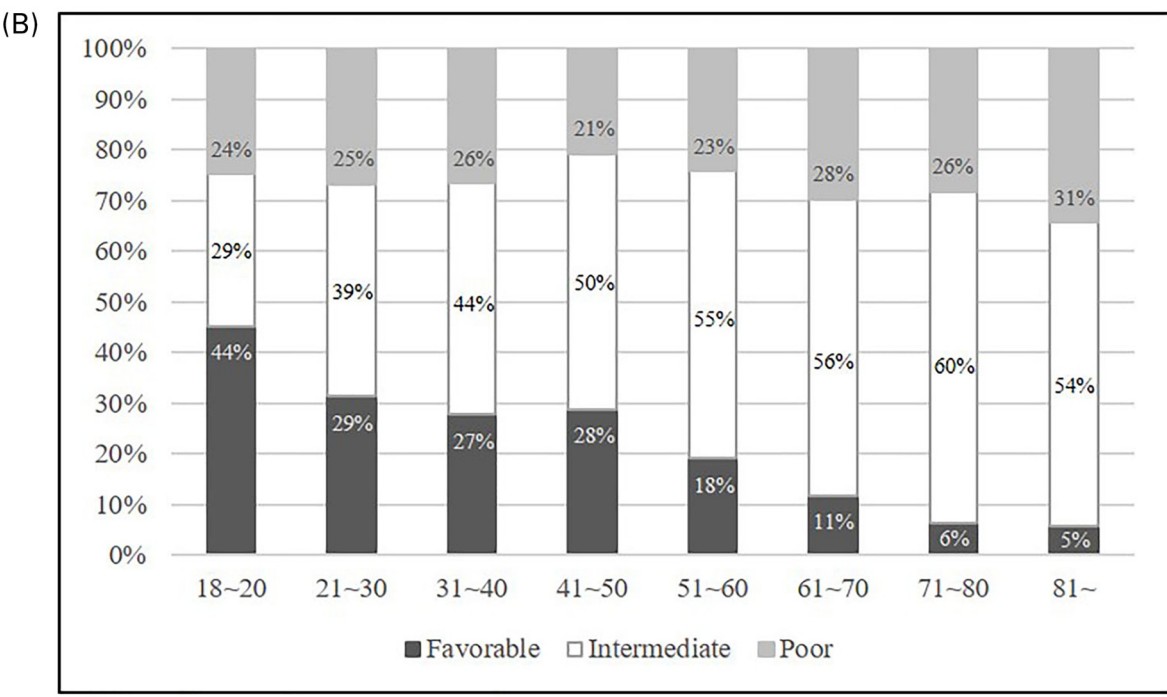

* Excluding unknown cytogenetics (n = 123)

**Fig 2. Patients stratified by age group (A) and cytogenetic risk (B).**

(A)
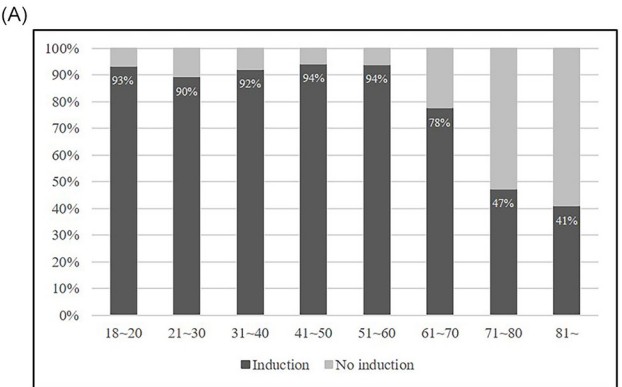

(B)
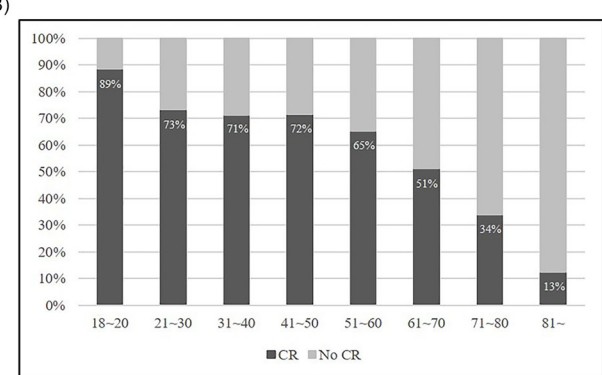

(C)
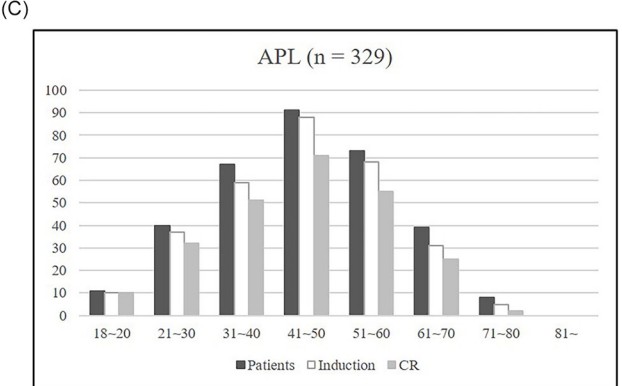

(D)
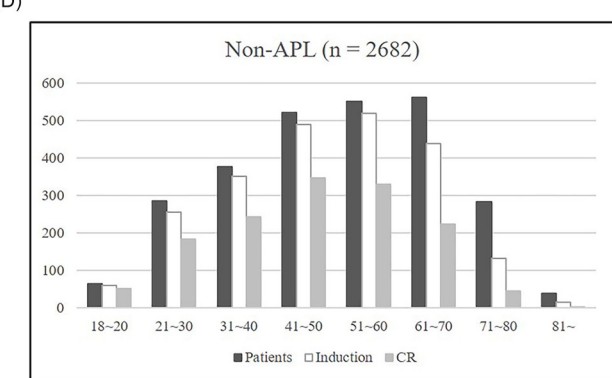

**Fig 3. Proportions of patients who received remission induction chemotherapy (A) and who achieved complete remission (CR) (B) by age group (all patients); Acute promyelocytic leukemia (APL) patients (C) and non-APL patients (D) who received remission induction chemotherapy and who achieved CR by age group.**

FLT3-ITD mutations and were treated with the AD regimen (n = 34) also showed improved OS as compared with those receiving the AI regimen (n = 105) (p = 0.005). However, in patients without FLT3-ITD mutations, there was no statistically significant difference in the OS observed between the two regimens. In the intermediate-risk group, the AD regimen led to a better OS than the AI regimen, regardless of patients' FLT3-mutation status (S2 Fig).

## Discussion

The results of this cohort study revealed an impact of age on the biology of patients with AML and suggested that the role of induction remission chemotherapy varies by age group. As the age of patients with AML increased, those in the favorable-risk group decreased rapidly, especially among those older than 50 years of age. The proportion of patients who received induction remission chemotherapy decreased among those older than 60 years and the CR rates among patients in this population who still received induction therapy were similarly reduced. In the case of APL, meanwhile, the proportion of induction therapy and the CR rate remained high until 70 years of age. Among non-APL patients, the OS decreased with age and the survival curves were well separated by age group.

Previously, several registry-based studies have reported the existence of a strong relationship between age and the clinical characteristics of AML [3, 4, 10] and poor outcomes of older adults have been well described, regardless of intensive treatments or transplantation [11, 12].

(A)

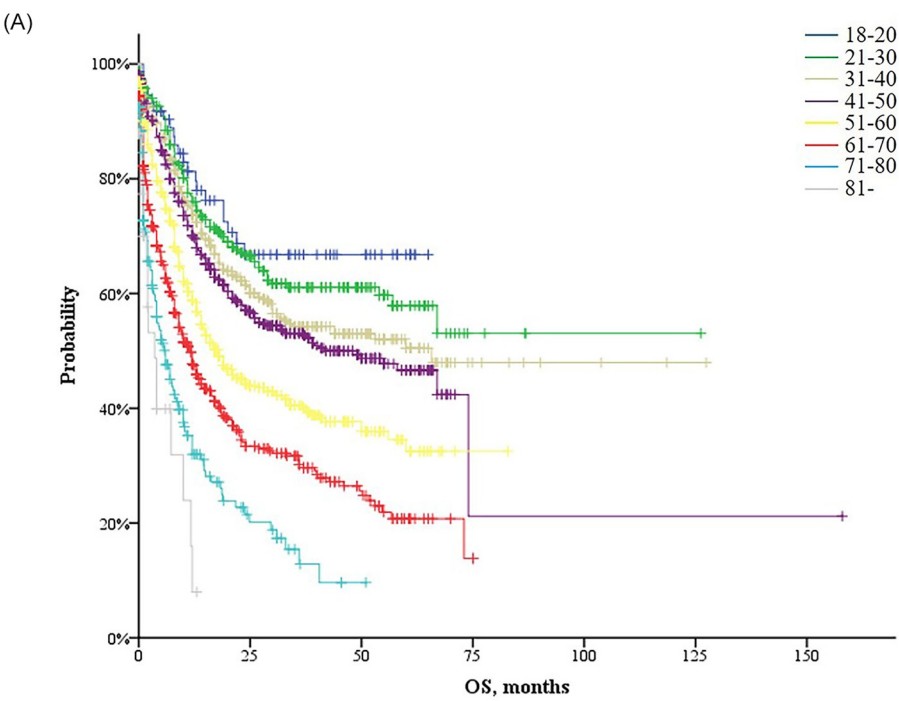

(B)

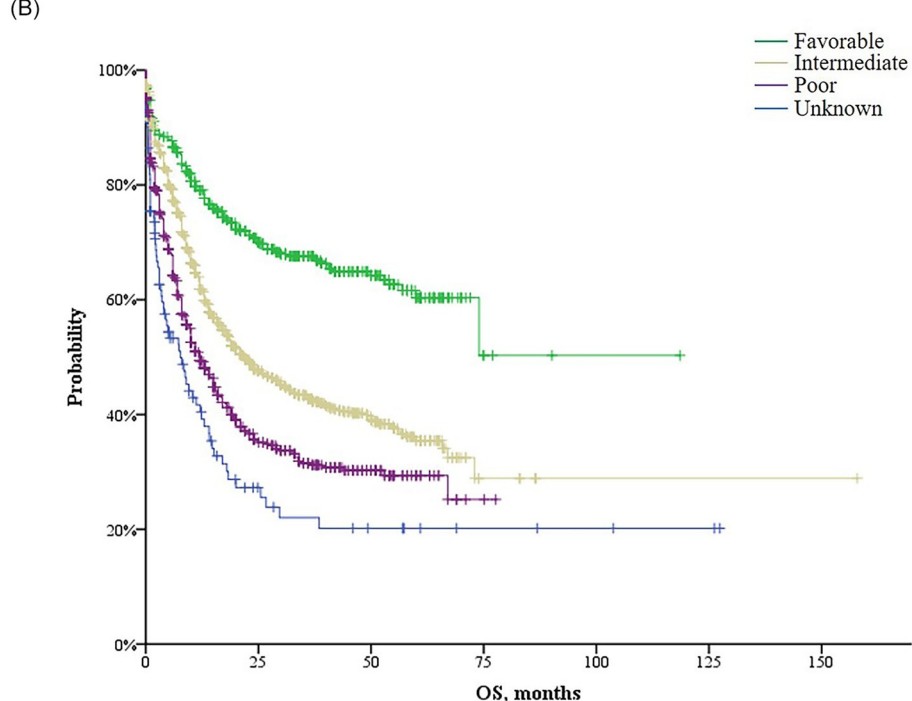

**Fig 4. Overall survival (OS) of all patients (n = 3,011) according to age group (A) and cytogenetic risk (B).**

**Table 2. Comparison of complete remission (CR) rates between patients receiving the cytarabine plus idarubicin (AI) and cytarabine plus daunorubicin (AD) treatment regimens according to age group.**

| Age group (years) | 18–20 | | 21–30 | | 31–40 | | 41–50 | | 51–60 | | 61–70 | | 71–80 | | ≥ 81 | | Total | |
|---|---|---|---|---|---|---|---|---|---|---|---|---|---|---|---|---|---|---|
| | (CR/total) | % | (CR/total) | % | (CR/total) | % | (CR/total) | % | (CR/total) | % | (CR/total) | % | (CR/total) | % | (CR/total) | % | (CR/total) | % |
| **AI** | 30/34 | 88.2 | 119/149 | 79.9 | 128/172 | 74.4 | 195/254 | 76.8 | 202/272 | 74.3 | 149/221 | 67.4 | 31/50 | 62.0 | 1/3 | 33.3 | 855/1155 | 74.0 |
| **AD** | 3/4 | 75.0 | 9/16 | 56.3 | 31/40 | 77.5 | 62/83 | 74.7 | 41/59 | 69.5 | 14/20 | 70.0 | 1/2 | 50.0 | 1/1 | 100.0 | 162/225 | 72.0 |
| **p-value** | 0.446 | | 0.053 | | 0.685 | | 0.700 | | 0.452 | | 0.813 | | 1.000 | | 1.000 | | 0.528 | |

AD: cytarabine plus daunorubicin, AI: cytarabine plus idarubicin.

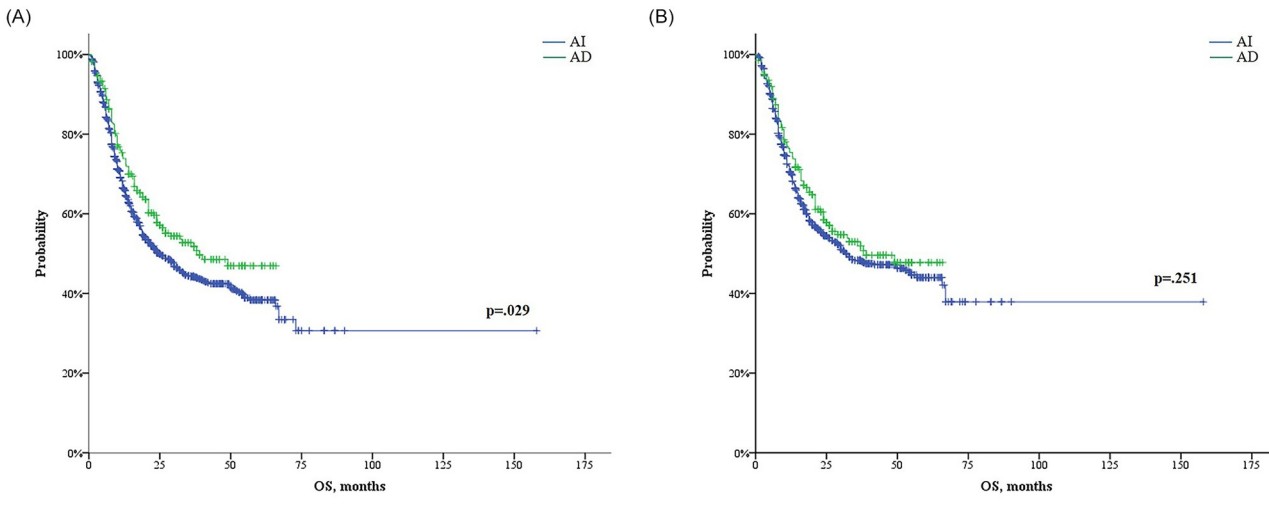

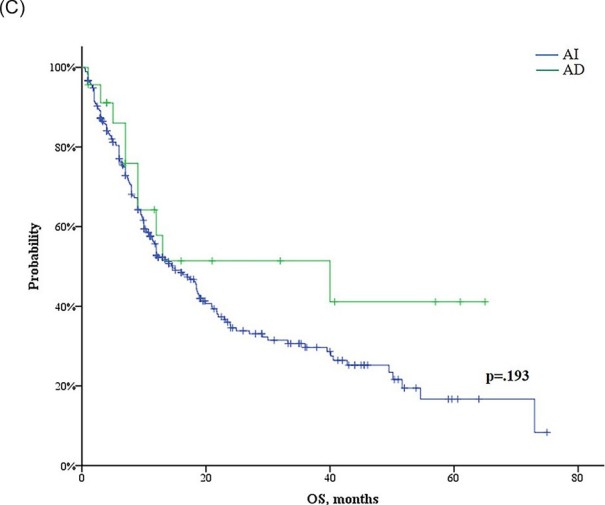

**Fig 5. Overall survival (OS) comparison of cytarabine plus idarubicin (n = 1,155) and cytarabine plus daunorubicin (n = 255) in all non–acute promyelocytic leukemia patients (A), those 60 years or younger (B), and those older than 60 years (C).**

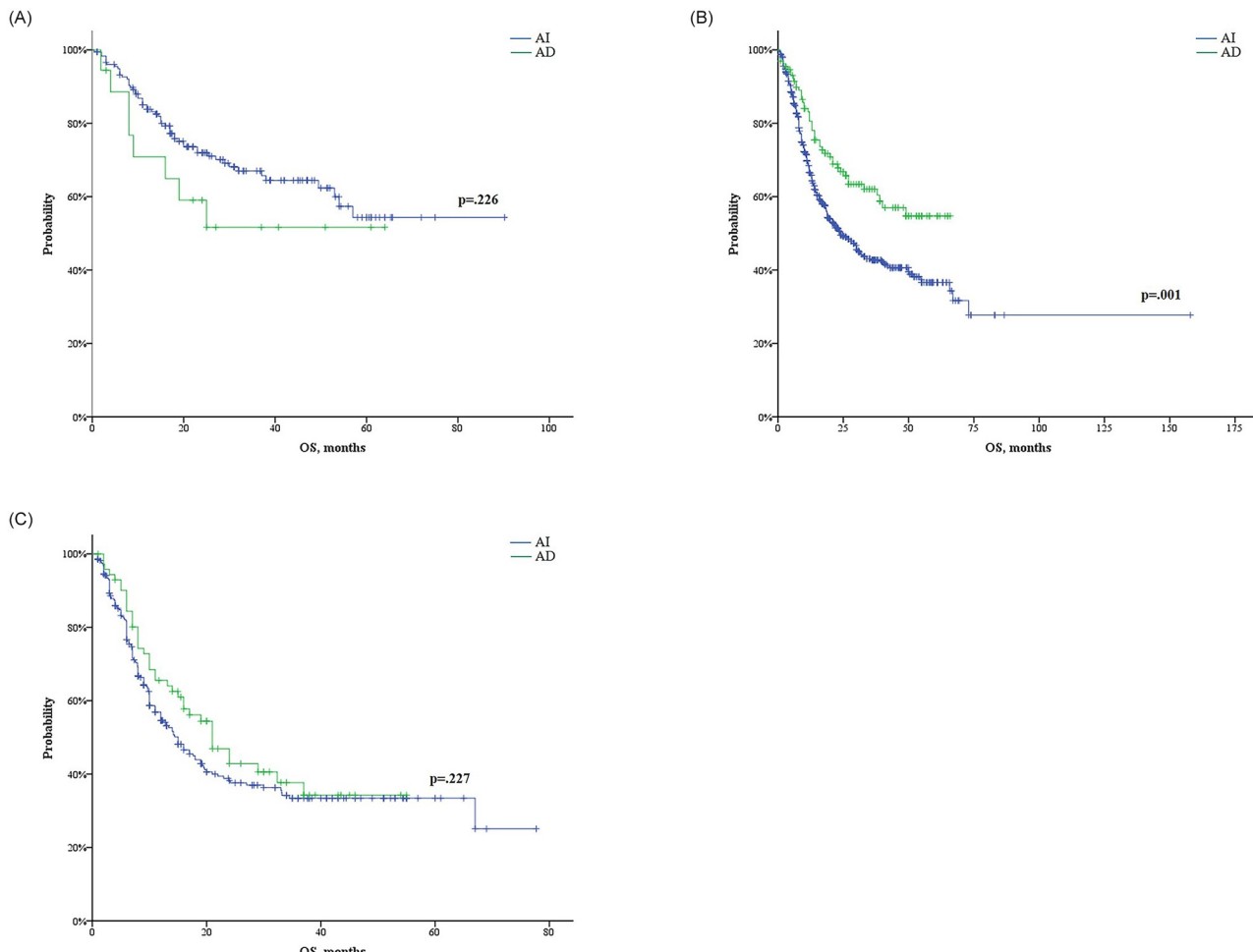

**Fig 6. Overall survival (OS) comparison of cytarabine plus idarubicin and cytarabine plus daunorubicin (AD) by cytogenetic risk: Favorable risk (A), intermediate risk (B), and poor risk (C).**

Some potent factors including decreased performance status, comorbidities, and different compositions of cytogenetic risk are considered to correlate with worse outcomes of AML in conjunction with aging [4, 13]. Treatment-related mortality is a major challenge in older and frail patients who receive induction chemotherapy [14, 15]. Despite using a granulocyte colony-stimulating factor (G-CSF) to shorten the duration of neutropenia, infection is the most common cause of death during induction therapy in this population [16, 17]. However, even with this explanation, the strikingly different outcomes among elderly patients, especially in those older than 60 years, are still not fully understood. In recent years, advanced genomic discovery studies have focused on AML and other myeloid neoplasms and have broadened understanding of the genetic landscape of AML and its impact on the pathophysiology of the disease [18–20]. These studies identically have pointed to the age-related frequency of recurring genetic mutations and several mutations were noted to have remarkably different incidence rates according to age [21, 22]. Most recently, the genetics of AML patients aged 75 years or older were reported and the clinical impact of intensive therapy was also analyzed [23]. AML in the older population exhibits intrinsic resistance to chemotherapy; moreover, these patients also show a higher proportion of adverse cytogenetic abnormalities and a greater

frequency of mutations associated with antecedent myeloid disease [24]. This study revealed adverse-risk cytogenetics to be a significant predictor for survival in intensively treated older AML patients.

In addition to the identification of genetic differences according to age, geographic and ethnic disparities in the clinical and genetic profiles of AML patients also exist [25–27]. An ethnic difference was also found in the context of APL [28]. One previous study reported different incidence rates of specific mutations existed in Chinese patients with AML relative to those observed in the Caucasian population [25], including lower incidence rates of FLT3-ITD and the DNMT3a mutation and a higher incidence of CEBPA in the former. Separately, Koh et al. previously reported the genomic signatures of Korean AML patients [29] by conducting whole-exome sequencing of AML samples from 103 patients and comparing the results with those of the Cancer Genome Atlas [18]. Interestingly, some mutations (TMPRSS13 and the TNN mutation) were uniquely found in Korean AML patients, while the frequencies of IDH2, NRAS, and DNMT3A were relatively low in this patient group. The evidence of ethnic differences is not conclusive yet and further studies involving larger numbers of patients are needed.

Using the same registry as adopted in our study, cytogenetic and molecular analyses were reported in 2016 [30]. This study was also an observational study focusing on the categorization of AML patients using conventional basic techniques. However, it revealed the geographic heterogeneity of Korean patients with AML and also reported cytogenetic data and their relationship with age. Among these data, while normal cytogenetics increased with age, favorable-risk cytogenetics such as t(15;17), t(8;21), and inv(16) were decreased in older age groups. Other cytogenetics associated with poorer outcomes including -5/del(5), -7/del(7), and FLT3-ITD mutations were also found to be more frequent in patients older than 50 years.

Besides analyzing the impact of age on the survival outcome, we also assessed the efficacy of different induction regimens, especially focusing on anthracyclines. An intravenous combination of anthracycline given for three days and cytarabine given for seven days has remained widely used since the 1980s [31–33]. In our study, AI was 4.5 times more frequently used than AD during induction remission chemotherapy of non-APL patients (n = 1,155 vs. n = 255). In comparing the efficacy of these two regimens, no difference in the CR rate, even during subgroup analysis by age or cytogenetic risk, was apparent. However, the AD regimen showed superior outcomes to those of the AI regimen in OS (39.0 vs. 24.8 months; p = 0.029). For almost 30 years, comparative studies of the efficacy of different types and doses of anthracyclines have been conducted [31, 32, 34]. Representatively, a randomized study comparing 13 mg/m$^2$ of idarubicin with 45 mg/m$^2$ of daunorubicin reported a higher CR rate affiliated with the former in younger (18–50 years) patients, though these drugs had similar toxicities [33]. Subsequently, a dose-intensification study of daunorubicin was performed and higher-dose (90 mg/m$^2$) daunorubicin resulted in a higher rate of CR and improved OS as compared with the standard dose (45 mg/m$^2$) in young patients [35]. Our study incidentally reported improved OS was associated with the AD regimen, especially among patients with FLT3-ITD mutations. Most recently, a phase III study comparing idarubicin with high-dose daunorubicin in AML patients with FLT3-ITD mutations demonstrated improved OS and event-free survival in the cytarabine plus high-dose daunorubicin treatment arm [36].

Our study derived data from a nationwide retrospective registry, which included limited clinical and cytogenetic information. Of note, different diagnostic methods were used for the clinical or cytogenetic categorization of AML during the period of registration. The lack of information regarding hematopoietic cell transplantation was a weakness of this study. We could not fully explain the mechanism associated with worsening outcomes linked with aging nor the impact of induction regimen on patients with different mutational characteristics. Despite these limitations, this is the first nationwide registry study performed in Korea

including more than 3,000 patients and it revealed real-world phenomena among Korean AML patients. In the future, more systemic prospective registry studies including extensive genetic profiling are necessary for the accurate characterization of AML patients in Korea.

## Supporting information

**S1 Fig. Overall survival (OS) curves of non–acute promyelocytic leukemia (APL) patients by age group (A), cytogenetic risk (B), and versus APL (C) patients.**
(TIF)

**S2 Fig. Overall survival (OS) curves of non–acute promyelocytic leukemia patients with (A) or without (B) FLT3-ITD mutations and intermediate-risk patients with (C) or without (D) FLT3-ITD mutations.**
(TIF)

## Acknowledgments

This research was conducted under the auspices of the acute myeloid leukemia/myelodysplastic syndrome (AML/MDS) working party of the Korean Society of Hematology (KSH). We thank the centers that participated in this study.

## Author Contributions

**Conceptualization:** Je-Hwan Lee, Hawk Kim.

**Data curation:** Hyeoung-Joon Kim, Yoo Hong Min, Dae-Sik Hong, Won Sik Lee, Hee-Je Kim, Ho-Jin Shin, Yong Park, Je-Hwan Lee, Hawk Kim.

**Formal analysis:** Kwai Han Yoo, Hawk Kim.

**Funding acquisition:** Je-Hwan Lee.

**Writing – original draft:** Kwai Han Yoo.

**Writing – review & editing:** Kwai Han Yoo, Hawk Kim.

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
