## [Decision Letter · Decision Letter 0]

28 Jan 2021

PONE-D-20-38695

Age and remission induction therapy for acute myeloid leukemia: an analysis of data from the Korean Acute Myeloid Leukemia Registry

PLOS ONE

Dear Dr. Kim,

Thank you for submitting your manuscript to PLOS ONE. After careful consideration, we feel that it has merit but does not fully meet PLOS ONE’s publication criteria as it currently stands. Therefore, we invite you to submit a revised version of the manuscript that addresses the points raised during the review process by both Reviewers, experts in the field.

We look forward to receiving your revised manuscript.

Kind regards,

Francesco Bertolini, MD, PhD

Academic Editor

PLOS ONE

Reviewers' comments:

Reviewer's Responses to Questions

**Comments to the Author**

1. Is the manuscript technically sound, and do the data support the conclusions?

Reviewer #1: No

Reviewer #2: Yes

2. Has the statistical analysis been performed appropriately and rigorously? 

Reviewer #1: Yes

Reviewer #2: Yes

3. Have the authors made all data underlying the findings in their manuscript fully available?

Reviewer #1: No

Reviewer #2: Yes

4. Is the manuscript presented in an intelligible fashion and written in standard English?

Reviewer #1: Yes

Reviewer #2: Yes

5. Review Comments to the Author

Reviewer #1: The authors have performed a retrospective analysis aiming to evaluate clinical outcomes of AML patients.

The paper is written in a good but essential English, and it could be clear for readers.

However the analysis is based on a old era of AML, considering that molecular alterations are not considered in the analysis.

Patients were collected between 2007 and 2011, and the paper is not applicable to current era of this disease, and also references confirm that paper has been written and thought not recently (last references are from 2018).

The idea could be good, considering the number of patients (3011), but it should be updated to recent data and recent prognostic parameters.

To date, it should be completely re-designed and rewritten, and it must be rejected at this state.

Reviewer #2: The authors present a large national cohort of AML patients from South Korea. In this article, they focus mainly on the following aspects: Changes in prognosis and cytogenetics with age, with the data nicely presented by decade. Treatment of APL vs non-APL patients, and outcomes with daunorubicin/Ara-C vs idarubicin/Ara-C. The discussion is in depth and develops the data findings.

A few points could be addressed in the manuscript:

-How were the 500 or so patients who did not receive intensive chemotherapy treated? What was their outcome?

-Is the registry collecting functional status and comorbidity data as, as the authors acknowledge themselves, these may contribute to age differences in prognosis?

6. PLOS authors have the option to publish the peer review history of their article (what does this mean?). If published, this will include your full peer review and any attached files.

Reviewer #1: No

Reviewer #2: **Yes: **Martine Extermann, MD, PhD

---

## [Author Response · Author response to Decision Letter 0]

9 Feb 2021

Reviewer #1: The authors have performed a retrospective analysis aiming to evaluate clinical outcomes of AML patients.

The paper is written in a good but essential English, and it could be clear for readers.

However the analysis is based on a old era of AML, considering that molecular alterations are not considered in the analysis.

Patients were collected between 2007 and 2011, and the paper is not applicable to current era of this disease, and also references confirm that paper has been written and thought not recently (last references are from 2018).

The idea could be good, considering the number of patients (3011), but it should be updated to recent data and recent prognostic parameters.

To date, it should be completely re-designed and rewritten, and it must be rejected at this state.

 I’d like to thank for the thorough review and inspection. Our study was based on the results of the registry program performed between 2007 and 2011 with the initiative of the AML/MDS working party of Korean Society of Hematology. As the reviewer pointed out, consistent molecular-based biological analysis was not possible because of the nature of a retrospective study and target population. New prospective cohort studies with molecular basis are needed.

Reviewer #2: The authors present a large national cohort of AML patients from South Korea. In this article, they focus mainly on the following aspects: Changes in prognosis and cytogenetics with age, with the data nicely presented by decade. Treatment of APL vs non-APL patients, and outcomes with daunorubicin/Ara-C vs idarubicin/Ara-C. The discussion is in depth and develops the data findings.

A few points could be addressed in the manuscript:

-How were the 500 or so patients who did not receive intensive chemotherapy treated? What was their outcome?

 Most of patients who did not received intensive chemotherapy were reported to have been transferred to another hospital. They were likely to have had intensive chemotherapy at the hospital they moved to. Due to the limitation of a registry study, the exact outcome of these patients was not known. Some elderly patients were reported to have given up intensive chemotherapy and just received palliative care.

-Is the registry collecting functional status and comorbidity data as, as the authors acknowledge themselves, these may contribute to age differences in prognosis?

 The performance status and comorbidities at the time of diagnosis are important to expect treatment outcome of acute leukemia. However, unified indexes for performance status and comorbidities were not applicable to this registry. Instead of these indicators, we focused on the patients who received induction chemotherapy, regarded as having qualified functional status.

---

## [Decision Letter · Decision Letter 1]

19 Apr 2021

Age and remission induction therapy for acute myeloid leukemia: an analysis of data from the Korean Acute Myeloid Leukemia Registry

PONE-D-20-38695R1

Dear Dr. Kim,

We’re pleased to inform you that your manuscript has been judged scientifically suitable for publication and will be formally accepted for publication once it meets all outstanding technical requirements.

As Reviewer #2 was no longer available, I have personally checked your answers to her/his issues, and found them adequate.

Kind regards,

Francesco Bertolini, MD, PhD

Academic Editor

PLOS ONE

Additional Editor Comments (optional):

Reviewers' comments:

Reviewer's Responses to Questions

**Comments to the Author**

1. If the authors have adequately addressed your comments raised in a previous round of review and you feel that this manuscript is now acceptable for publication, you may indicate that here to bypass the “Comments to the Author” section, enter your conflict of interest statement in the “Confidential to Editor” section, and submit your "Accept" recommendation.

Reviewer #1: All comments have been addressed

2. Is the manuscript technically sound, and do the data support the conclusions?

Reviewer #1: Yes

3. Has the statistical analysis been performed appropriately and rigorously? 

Reviewer #1: Yes

4. Have the authors made all data underlying the findings in their manuscript fully available?

Reviewer #1: Yes

5. Is the manuscript presented in an intelligible fashion and written in standard English?

Reviewer #1: Yes

6. Review Comments to the Author

Reviewer #1: (No Response)

7. PLOS authors have the option to publish the peer review history of their article (what does this mean?). If published, this will include your full peer review and any attached files.

Reviewer #1: **Yes: **Claudio Cerchione, MD, PhD - Hematology Unit, Istituto Scientifico Romagnolo per lo Studio e la Cura dei Tumori (IRST) IRCCS, Via Piero Maroncelli 40, 47014, Meldola (FC), Italy.

---

## [Editor Report · Acceptance letter]

22 Apr 2021

PONE-D-20-38695R1 

Age and remission induction therapy for acute myeloid leukemia: an analysis of data from the Korean Acute Myeloid Leukemia Registry 

Dear Dr. Kim:

I'm pleased to inform you that your manuscript has been deemed suitable for publication in PLOS ONE. Congratulations! Your manuscript is now with our production department. 

Kind regards, 

on behalf of

Dr. Francesco Bertolini 

Academic Editor

PLOS ONE